# Methodological Quality Assessment with the AGREE II Scale and a Comparison of European and American Guidelines for the Treatment of Lyme Borreliosis: A Systematic Review

**DOI:** 10.3390/pathogens10080972

**Published:** 2021-07-31

**Authors:** Steve Nguala, Elisabeth Baux, Solène Patrat-Delon, Florian Saunier, Julien Schemoul, Pierre Tattevin, Céline Cazorla, Carole Eldin, Kevin Bouiller, Alice Raffetin

**Affiliations:** 1Department of Infectious Diseases, Tick-Borne Diseases Reference Center, North Region, Hospital of Villeneuve–Saint Georges, 40 Allée de la Source, 94190 Villeneuve-Saint-Georges, France; steve.nguala@chiv.fr; 2Department of Infectious Diseases, Tick-Borne Diseases Reference Center, East Region, Hospitals of Brabois—University Hospital of Nancy, Rue du Morvan, 54500 Vandoeuvre Les Nancy, France; e.baux@chru-nancy.fr; 3Department of Infectious Diseases and ICU, Tick-Borne Diseases Reference Center, West Region, University Hospital of Rennes Pontchaillou, CHU Rennes Pontchaillou, 2 Rue Henri Le Guilloux, 35033 Rennes, France; solene.patrat-delon@chu-rennes.fr (S.P.-D.); pierre.tattevin@chu-rennes.fr (P.T.); 4Department of Infectious Diseases, Tick-Borne Diseases Reference Center, Central Region, University Hospital of Saint-Etienne, Avenue Albert-Raimond, 42270 Saint-Priest-en-Jarez, France; florian.saunier2@etu.univ-st-etienne.fr (F.S.); celine.cazorla@chu-st-etienne.fr (C.C.); 5Department of Rheumatology, Tick-Borne Diseases Reference Center, North Region, Hospital of Villeneuve-Saint Georges, 40 Allée de la Source, 94190 Villeneuve-Saint-Georges, France; julien.schemoul@chiv.fr; 6Tick-Borne Diseases Reference Center, South Region IHU-Méditerranée Infection, 13005 Marseille, France; carole.eldin@ap-hm.fr; 7IRD, AP-HM, SSA, VITROME, Aix Marseille University, 13005 Marseille, France; 8Department of Infectious Diseases, University Hospital of Besançon, 3 Boulevard Alexandre Fleming, 25000 Besançon, France; kbouiller@chu-besancon.fr; 9UMR CNRS 6249 Chrono-Environnement, University of Franche-Comte, 25000 Besancon, France; 10European Study Group for Lyme Borreliosis ESGBOR, ESCMID, Gerbergasse 14 3rd Floor, 4001 Basel, Switzerland

**Keywords:** Lyme borreliosis, treatment, guidelines, AGREE II

## Abstract

Background: Most European and American countries recently updated their guidelines on Lyme borreliosis (LB). The aim of this study was to provide a comparative overview of existing guidelines on the treatment of LB in Europe and America and to assess the methodological quality of their elaboration. Methods: A systematic search was carried out in MEDLINE, Google Scholar, and the national databases of scientific societies from 2014 to 2020. Quality was assessed by two independent reviewers using the Appraisal of Guidelines for Research and Evaluation II (AGREE II) tool. Results: Twelve guidelines were included. The scores for the AGREE II domains (median ± IQR) were: overall assessment 100 ± 22, scope and purpose 85 ± 46, stakeholder involvement 88 ± 48, rigour of development 67 ± 35, clarity of presentation 81 ± 36, applicability 73 ± 52 and editorial independence 79% ± 54%. Cohen’s weighted kappa showed a high agreement (K = 0.90, 95%CI 0.84–0.96). Guidelines were quite homogeneous regarding the recommended molecules (mostly doxycycline in the first intention and ceftriaxone in the second intention), their duration (10 to 28 days), and their dosage. The differences were due to the lack of well-conducted comparative trials. The International Lyme and Associated Diseases Society (ILADS) guidelines were the only ones to suggest longer antibiotics based on an expert consensus. Conclusion: European and American guidelines for the treatment of LB were quite homogeneous but based on moderate- to low-evidence studies. Well-conducted comparative trials are needed to assess the best molecules, the optimal duration and the most effective doses.

## 1. Introduction

Lyme borreliosis (LB) is the most frequent vector-borne disease in Europe and the USA [1,2]. The number of cases in Europe has increased steadily over the last two decades. The incidence of LB varies from a country to another from 0.001/100,000 to 464/100,000, with an unweighted mean incidence rate of 56.3/100,000 persons per year (∼232,125 cases/year) [1]. LB is transmitted by ticks and caused by spirochetes of the *Borrelia burgdorferi* sensu lato complex. The most frequent clinical pictures in Europe are erythema migrans (EM) and Lyme neuroborreliosis (LNB), but it can also affect joints such as Lyme arthritis (LA), heart and eyes [3,4]. The diversity of the clinical manifestations in Europe is due to the larger genospecies of *Borrelia burgdorferi* sensu lato (*B. afzelii, B. garinii, B. burgdorferi sensu stricto* etc.) compared to those in the USA (*B. burgdorferi* sensu stricto) [3]. Microbiological diagnosis mainly relies on a two-tier serological test and PCR according to the stage of the disease and the anatomical site sample [5,6,7,8]. An antibiotic therapy among doxycycline or beta-lactamin (ceftriaxone or amoxicillin) or azithromycin is prescribed between 10 and 28 days, also according to the stage of the disease (early < 6 months or late > 6 months) and the clinical manifestation. For the early localized stage (EM), antibiotic duration varies from 10 to 21 days, and for the disseminated stages, it ranges from 14 to 28 days. Long-term antibiotics (more than three months) have not shown superiority in any randomized trials [9,10,11,12,13]. In vitro antibiotic resistance has not been observed [14,15,16]. Some subjective symptoms may exist at all the stages and may persist after a well-conducted antibiotic therapy. This is called the post-treatment Lyme disease syndrome (PTLDS). Antibiotics are usually not recommended for the latter because there is no scientific evidence for the persistence of an active *Borrelia*, in clinically relevant studies [7].

Most of the European guidelines for the diagnosis and management of LB have been updated recently. They are consistent for the clinical and microbiological diagnosis of LB [6], but there are disparities between the recommended molecules, their duration and their dosage.

The aim of this study was to provide a comparative overview of the existing guidelines on the treatment of LB in Europe and America and to assess the methodological quality of their elaboration.

## 2. Results

The search strategies identified four guidelines from MEDLINE [7,17,18,19], two from Google Scholar [20,21], one from the Haute Autorité de Santé (HAS) [22], one from the National Institute for Health and Care Excellence (NICE) [23], two from.

The Arbeitsgemeinschaft der Wissenschaftlichen Medizinischen Fachgesellschaften (AWMF) [24,25], one from the Belgian Antibiotic Policy Coordination Committee (BAPCOC) [26] and one from the International Lyme and Associated Diseases Society (ILADS) [27]. The remaining number of guidelines found in each database after removal of duplicates and then selected for further review and met the inclusion criteria are shown in Figure 1. Eleven guidelines were developed by official academic societies [7,17,18,19,20,21,22,23,24,25,26] and one by a non-official organisation, the ILADS, defined as a ‘transdisciplinary medical association’ of physicians and researchers working on Lyme and tick-borne diseases (Table 1) [27]. This association is not officially recognised by American authorities as an academic society.

### 2.1. Methodological Quality Assessment (cf. Table 2)

#### 2.1.1. Overall Assessment 

The overall quality of guidelines varied, but most of the methodological quality of guidelines was admissible. The overall quality was more than 50% in nine included guidelines [7,17,18,19,22,23,24,25,26], especially in five [7,19,23,24,25] which scored 100%. Two guidelines relied on the GRADE approach [19,27], which is the highest quality standard. Six guidelines were rated according to AGREE II as ‘recommended by reviewers’ (score > 80%) [7,17,19,23,24,25], three as ‘recommended with modifications’ (score between 50% and 80%) [18,22,26] and three as ‘not recommended’ (score < 50%) [20,21,27]. Cohen’s weighted kappa showed that appraisers reached a high agreement (K = 0.90, 95%CI 0.84–0.96) (Table 2).

#### 2.1.2. Scope and Purpose 

Overall objectives were described in eleven guidelines [7,17,18,19,20,22,23,24,25,26,27], and specific health questions were described in eight [7,17,19,22,23,24,25,27]. The population to whom the guidelines were meant to apply were exactly described in ten [7,17,18,19,20,22,23,24,25,27].

#### 2.1.3. Stakeholder Involvement 

The domain stakeholder involvement was described to some degree. Seven guidelines provided their development group [7,19,22,23,24,25,27], seven attempted to find the views and preferences of the target population [7,19,22,23,24,25,27], and nine clearly defined target users [7,17,18,19,22,23,24,25,27].

#### 2.1.4. Rigour of Development 

Systematic methods to search for evidence were reported in nine guidelines [7,17,18,19,22,23,24,25,27]. Criteria for selecting evidence and method for formulating the recommendation were described in six guidelines [7,19,22,23,24,25]. Seven described the strengths and limitations [7,17,19,22,24,25]. Seven considered the health benefits, side effects and risks in formulating the recommendations [7,17,18,19,22,24,25]. Eight presented the link between recommendations and supporting evidence as reference to the respective evidence [7,17,18,19,22,23,24,25]. Seven were externally reviewed by experts before their publications [7,19,22,23,24,25,26]. Only one provided an updating procedure [22].

#### 2.1.5. Clarity of Presentation 

Nine guidelines had specific and unambiguous recommendations [7,17,18,19,22,23,24,25,26]. Nine provided different options for the management of conditions or health issues [7,17,18,19,22,23,24,25,26]. All guidelines identified key recommendations [7,17,18,19,20,21,22,23,24,25,26,27].

#### 2.1.6. Applicability 

Eleven guidelines described facilitators and barriers to applications [7,17,18,19,20,21,22,23,24,25,26]. All guidelines provided advice and/or tools on how recommendations can be put in practice to some degree [7,17,18,19,20,21,22,23,24,25,26,27]. Eight considered the potential resource implications of applying the recommendations [7,17,19,21,22,23,24,25], and a further eight presented monitoring and/or auditing criteria, including patients’ advice [7,17,18,19,22,23,25,26].

#### 2.1.7. Editorial Independence 

Editorial independence was evaluated by two parameters: “The views of the funding body have not influenced the content of the guideline” and “Competing interests of guideline development group members have been recorded and addressed”. Eight guidelines explained that the views of the funding body had no influence on content [7,17,18,19,23,24,25,26]. Conflicts of interest were revealed in nine guidelines [7,17,18,19,22,24,25,26], but the mechanism of these conflicts’ potential influence on recommendations was unclear.

### 2.2. Comparison of Recommended LB Treatments

#### 2.2.1. Treatment of Skin Manifestations (cf. Table 3)

##### Erythema Migrans (EM)

Of the 12 guidelines, 2 did not recommend a specific treatment for EM because they dealt with LNB and PTLDS only [17,25] (Table 3).

The Belgian guidelines were based on 7 articles [26], the German ones on 1 article and on American and European guidelines [24], the European ones on 7 articles [18], and the Canadian ones cited a previous Canadian guideline [21]. Only three guidelines were elaborated according to the Grading of Recommendations Assessment, Development, and Evaluation (GRADE) approach [7,19,27]: the French scientific societies’ guidelines were based on 4 articles that evaluated the recommendation as grade B [7], the IDSA guidelines were based on 41 references that evaluated the recommendations as possessing strong to moderate quality of evidence [19], and the ILADS guidelines were based on 9 references that evaluated the available studies as having low evidence for their recommendation [27]. Despite the analysis of common articles [28,29,30,31,32,33,34,35,36] and the same methodology (GRADE) the conclusions of these three guidelines were widely different [7,19,27]. The reason lies in the interpretation of the literature. 

The two main cited articles were both randomized double-blind trials [32,37]. In the first, 246 adult patients with EM received amoxicillin 500 mg × 3/d for 20 days or azithromycin 500 mg/d and a placebo for 7 days. Patients treated with amoxicillin were significantly more likely than those treated with azithromycin to achieve complete resolution of disease at day 20 (*p* = 0.024) and had less relapse (*p* = 0.005). The authors concluded that a 20-day course of amoxicillin was effective for EM [32]. The second trial included 180 patients with EM and compared 10 days of oral doxycycline with or without a single intravenous dose of ceftriaxone or 20 days of oral doxycycline. In the on-study analysis, the complete response rate at 30 months was 83.9% in the 20-day doxycycline group, 90.3% in the 10-day doxycycline group and 86.5% in the doxycycline-ceftriaxone group (*p* > 0.2). There were no significant differences in the results of neurocognitive testing among the three treatment groups and a control group without LB. Diarrhea occurred significantly more often in the doxycycline-ceftriaxone group (35%) (*p* < 0.001). The authors concluded that extending treatment with doxycycline from 10 to 20 days or adding one dose of ceftriaxone to a 10-day course of doxycycline did not enhance therapeutic efficacy in patients with EM [37].

While amoxicillin (1.5 to 3 g/d) and doxycycline (200 mg/d) were recommended for EM treatment in 10/10 guidelines, cefuroxime-axetil (500 mg × 2) was recommended for 14 to 28 days in 6/10 [19,20,24,26,38] and for 4 to 6 weeks in 1/10 [27]. Azithromycin was also recommended with various doses and various lengths (5–10 days) in 5/10 guidelines [18,19,22,24,26]. Clarithromycin and phenoxymethylpenicillin were only recommended in one guideline each [18,26].

Evidence shows no clear difference between doxycycline, amoxicillin and azithromycin in patients with EM except for one trial [32]. Therefore, azithromycin is sometimes considered as a second-line agent, with some efficacy concerns. Three guidelines [18,19,26] underscored this discordant result for azithromycin without a clear explanation apart from methodological differences; other guidelines considered azithromycin to be adequately effective without restriction or discussion [23,24,27].

The recommended length of treatment varies from 10 to 14 days in most guidelines [7,18,19,22,26], but it was increased to 21 [23], or 28 days [20] or 4–6 weeks [27] in 3 outsiders. To assess the need for prolonged treatment, the ILADS working group performed a Medline search for prospective randomized clinical trials investigating the effectiveness of 5–20 days of oral azithromycin, cefuroxime, doxycycline, phenoxymethylpenicillin, or amoxicillin for the treatment of EM [27]. Of the 76 papers identified, 9 were evaluated [28,29,30,32,33,35,36,39]. The authors concluded that the evidence quality was low and insufficient to recommend a short treatment (<21 days) [27]. Nevertheless, no study was cited to assess the superiority or the non-inferiority of prolonged treatment because no randomized controlled trial explored treatment longer than 21 days [27].

##### Borrelial Lymphocytoma (BL)

Regarding BL, only 6/12 guidelines edited recommendations [7,18,19,20,22,24]. All these guidelines recommended the use of amoxicillin (1.5 to 3 g/d) or doxycycline (200 mg/d) in the first intention. The use of azithromycin with various doses was less consensual [18,22,24], and cefuroxime axetil (500 mg × 2/d) was recommended in only 4/6 [7,20,22,24].

Treatment duration for BL with amoxicillin or doxycycline varied from 14 to 28 days, mostly 14 to 21 days in 4/6 guidelines [7,20,22,24]. These discrepancies are probably due to the absence of available randomized controlled trials. German Dermatology Society guidelines and the two French guidelines provided a duration of 21 days because BL indicates a disseminated LB [7,22,24] whereas no explanation on duration choice is given for other guidelines [10,11,12]. The recommendations levels were weak, when specified (grade B for French scientific societies guidelines and weak recommendation of low quality for IDSA), based on extremely few references. Only a case report [39] and 4 cohorts assessed these choices.

##### Acrodermatitis Chronica Atrophicans (ACA)

All but four guidelines provided recommendations for the treatment of ACA [7,18,19,20,22,23,24,26]. Because no randomized controlled trial for treatment duration for ACA was published, guidelines were based on cohort studies [38,40,41,42,43,44], with low quality of evidence, mainly because of the low number of patients.

The recommended duration was homogenous, longer than 21 days [7,18,19,22,23,24,26], except for Polish guidelines [20] without any supplementary explanation. In the same way, recommendations for antibiotic choice were homogeneous. Doxycycline (200 mg/d) was recommended to treat ACA in all guidelines [7,18,19,20,22,23,24,26] and ceftriaxone (2 g/d) in 6/8 guidelines [7,20,22,23,26], specifically if associated with neurological symptoms in a German Dermatology Society guideline [24]. Amoxicillin (1.5 to 3 g/d) was recommended in 5/8 [18,19,20,23,24] and cefuroxime axetil (500 mg × 2) in 3/8 guidelines [18,19,20].

##### Children and Pregnant Women

Guidelines in children were mentioned in 9/10 guidelines [7,18,19,21,22,23,24,26,27]. For EM treatment in children under eight years old, doxycycline is avoided based on experience with older tetracyclines (staining of permanent teeth). This adverse effect has not been reported with doxycycline, and some studies reported its good tolerability in children for a duration of up to 14 days [25]. The recommended treatments followed the adult schemes in all guidelines [7,10,11,13,14,15,16,18,19] with dose adaptation. Amoxicillin was prescribed at 50 mg/kg/d in three divided doses [7,21,22,24,26,27] except in NICE guidelines in which amoxicillin was proposed at 30 mg/kg/d until 33kg [23]. When recommended, cefuroxime-axetil dose was 30 mg/kg/d in two divided doses [21,24,26,27]. Azithromycin dose varied according to recommendations: 10 mg/kg on day 1, then 5–10 mg/kg/d [27], 20 mg/kg on day 1 and 10 mg/kg/d for 4 days [26], 20 mg/kg/d for 4 days [7], and 10 mg/kg/d until 50 kg for 17 days [23]. In the French scientific societies guidelines, azithromycin was not recommended for adult patients but was possible for children under eight years old only in cases of allergy to beta-lactams (dose 20 mg/kg/d, max 500 mg/f, for 5 days) [7]. When children are over eight years old, the recommended dose for doxycycline is 4 mg/kg/d in two divided doses, with a maximum of 200 mg/d. Of note, the IDSA guidelines suggested further study to establish the safety profile of doxycycline in children but stated that data are reassuring on the safety of short courses of doxycycline [19].

Most guidelines raised awareness about the molecules when contraindicated in pregnancy [18,19,21,23,26]. Polish guidelines recommended the use of β-lactams for the treatment of BL in pregnant women [20]. Other recommendations were more specific: amoxicillin was the first line in all the recommendations, giving specific treatment for pregnant women [7,19,22,24]. Azithromycin was the second-line agent in the French High Health Authority guidelines [22], while German guidelines recommended as a second line Penicillin G (IV) or ceftriaxone (IV) [24]. Cefuroxime, ceftriaxone, cefotaxime or azithromycin were placed as the third line of treatment in these German guidelines [24].

#### 2.2.2. Treatment of Lyme Neuroborreliosis (LNB) (cf. Table 4)

Lyme neuroborreliosis (LNB) is the most common manifestation of disseminated LB in Europe. The two main molecules recommended by all guidelines were oral doxycycline and intravenous ceftriaxone. European studies have shown similar efficacy of doxycycline for the treatment of early LNB compared to parenteral ceftriaxone. Because the number of patients with severe LNB (encephalitis, encephalomyelitis) was limited in these studies, some guidelines recommended ceftriaxone as the first choice for the treatment of LNB with central nervous system (CNS) involvement pending more robust data [18,20,22,26]. However, complications related to intravenous treatment, such as thrombophlebitis and secondary infections, would be avoided by using oral antibiotics, also associated with lower costs, and a higher degree of patient satisfaction [45]. Therefore, most guidelines recommended the use of oral doxycycline rather than intravenous treatment [7,19,20,23,25,26]. NICE and IDSA guidelines recommended ceftriaxone as first-line therapy only for LNB with central nervous system impairment [19,23]. For the treatment of late LNB (evolution > 6 months), all guidelines recommended ceftriaxone as first-line therapy except the French scientific societies guidelines, which left the choice between doxycycline and ceftriaxone for the first line considering the explanations above [7]. Only NICE and IDSA guidelines did not specify therapy for late LNB but differentiated between CNS and peripheral nervous system (PNS) involvement [19,23]. Of note, only the IDSA and the German guidelines proposed the use of cefotaxime and Penicillin G for the treatment of LNB [19,25]. However, these two molecules are more difficult to manage compared to ceftriaxone because of shorter half-lives (Table 4).

Dosage of doxycycline was 100 mg × 2/day for all guidelines except for the German and IDSA guidelines, which recommended an increased dosage of 300 mg in 1 or 3 divided doses or 200 mg in 1 dose [19,25]. The French scientific societies suggested 200 mg × 2/d for CNS LNB [7]. Similarly, NICE guidelines recommended intravenous ceftriaxone 4 g/day as the initial treatment for adults and teenagers (age 12 and over) with LNB affecting the CNS whereas all other guidelines recommended 2 g/day [15]. These differences may be related to pharmacokinetics concerns and higher doses of ceftriaxone being routinely recommended for bacterial meningitis (i.e., 70–100 mg/kg/day). Doxycycline, with a daily dose of 200 mg, does not reach the minimal bactericidal and inhibitory concentrations for all strains of *B. burgdorferi* in the cerebrospinal fluid [14,46,47,48]. However, most prospective studies used a dosage of 200 mg/day [47,49,50]. One study used doxycycline 400 mg/day and another 200 mg for 2 days then 100 mg/day for the remaining 8 days [51,52]. No studies compared different dosages.

Treatment duration varied from 14 to 28 days in different guidelines depending on the type of neurologic impairment. These differences depended on the interpretation of the literature. Most studies analysed in the guidelines included patients with early LNB [52,53,54,55]. The duration of treatment in randomized controlled trials was mainly 10–14 days [52,53,54,55]. Results of two randomized controlled studies showed no benefit from an antibiotic therapy longer than 28 days [9,56]. However, Cadavid et al. reported in a systematic review a low to an extremely low quality of clinical evidence of studies about LNB because of the limited number and heterogeneous trials [57]. Kortela et al. reported, in a randomized, open-label trial, 210 adults with suspected LNB treated with oral doxycycline for 28 days or intravenous ceftriaxone for 21 days [58]. Late LNB represented 10% of cases. Patients improved equally well with both treatment regimens. Of note, there was no clear benefit from longer treatments, with a persistence of residual symptoms [58]. However, this study was too recent to be included in guidelines.

##### Children and Pregnant Women

There was limited evidence on the use of antibiotics to treat LNB in children, and available studies are of poor quality [59]. All guidelines recommended the same treatment for children aged 9 to 12 than for adults and teenagers but with doses adjusted by weight: doxycycline 2.5–5 mg/kg, ceftriaxone 50–80 mg/kg. The use of doxycycline in children under 9 years old is currently limited by licensing and clinical experience. Some guidelines discussed the place of doxycycline in children’s LNB, especially when beta-lactams are contraindicated or when the intravenous line is difficult to manage [7,19]. However, doxycycline may be prescribed in children younger than eight years old only after having informed the parents that such treatment does not have a marketing authorization for use in this population.

Five out of eight recommendations contraindicated doxycycline in pregnant women [7,20,22,23,25] because of the staining of permanent teeth in their babies. Based on experience with older tetracycline, not with doxycycline, IDSA guidelines specified that the safety of doxycycline in pregnancy and breastfeeding required more study, and thus, the decision to use doxycycline in these patients should be individualized to the likely risks and benefits of alternative antibiotics [27]. In the same way, Belgium’s recommendations contraindicated tetracycline in the second trimester of pregnancy but not doxycycline [26]. French recommendations indicated that doxycycline administered after the first trimester of pregnancy was associated with a risk of coloration of deciduous teeth, without any impact on permanent teeth [7,22]. However, they recommended amoxicillin or ceftriaxone as first-line treatment depending on the disease stage in pregnant women [7,22].

##### 2.2.3. Treatment of Lyme Arthritis (cf. Table 4)

Seven guidelines dealt with the treatment of LA [7,18,19,20,22,23,26]. There was no clear distinction made on the treatment of forms that had been evolving for less (early LA) or more (late LA) than six months in any guidelines [7,18,19,20,22,23,26].

Two leading articles were cited in 5/7 guidelines. Steere et al. (1985) demonstrated in a double-blind placebo-controlled trial in 40 patients with established LA that 2.4 million UI of intramuscular benzathine penicillin weekly for three weeks (n = 20) was superior to placebo (n = 20) with 35% favorable outcome soon afterwards (*p* < 0.02) [60]. This study was the first to show the efficacy of an antibiotherapy for LA, but the regimens that were tested were not uniformly effective, suggesting that further studies would be needed to determine the optimal course of therapy [60]. Steere et al. (1994) performed a randomized controlled trial comparing doxycycline or amoxicillin-probenecid and a placebo for 30 days in patients presenting LA, defined as clinical arthritis and positive serology for *Borrelia* [61]. Meanwhile, 18/20 patients and 16/20 patients respectively treated by doxycycline and amoxicillin-probenecid had their arthritis resolved within 1 to 3 months (complete answer to antibiotics at 48% vs. 0% for placebo, *p* = 0.02) [61]. Nonetheless, 5 patients developed LNB later, 4 of whom had received the amoxicillin-probenecid regimen [61]. Moreover, another group of 16 additional patients who had persistent arthritis for at least 3 months after treatment with oral antibiotics or parenteral penicillin was given intravenous ceftriaxone for 2 weeks, and none had resolution of arthritis within 3 months [61]. Authors concluded that LA could be treated by oral antibiotics, but patients may still develop LNB, and patients with certain genetic and immune markers may have persistent arthritis despite treatment with oral or intravenous antibiotics [61].

Two randomized controlled trials conducted by Dattwyler et al., cited in 4/7 guidelines, investigated (i) the efficacy of ceftriaxone vs. penicillin, and (ii) two durations of antibiotic therapy (14 days vs. 28 days) [62,63]. An extended duration of 28 days showed a better response without achieving statistical significance (*p* = 0.07) and presented more side effects (*p* < 0.02) [62]. Intravenous ceftriaxone seemed more effective than intravenous penicillin but with a small sample size (n = 23) [64]. In 31 patients, ceftriaxone 4 g/d was not superior to ceftriaxone 2 g/d [64].

Several cohort studies, cited in 2/7 guidelines, described the clinical evolution of patients treated for late LA, mainly by doxycycline, ceftriaxone, cefuroxime-axetil and amoxicillin [64,65,66,67]. A complete response was noted in 45% to 75% of cases after the first line of antibiotic therapy for 3–4 weeks and in 20% to 25% after a second line. Oral therapy presented fewer side effects.

All these studies enable us to conclude that several antibiotics are effective among doxycycline, ceftriaxone, and amoxicillin for LA, although data about their optimal doses and duration are limited. Ceftriaxone and doxycycline seemed both as effective as and more effective than amoxicillin or intravenous penicillin. Intravenous ceftriaxone presented more adverse events than oral therapy. The data suggested the superiority of a 28-day treatment compared to a 14-day treatment but with a low level of evidence. About a quarter of the patients presented refractory arthritis after the second line of antibiotics despite the absence of *Borrelia* in the articular liquid, suggesting an inflammatory process responding to anti-inflammatory therapies. Of note, no studies directly assessed the efficacy of cefuroxime-axetil versus other oral antibiotics or placebo in the treatment of LA. Evidence is inferred from studies of its efficacy in the treatment of early manifestations [19].

Doxycycline was recommended at the first line in all the guidelines [7,18,19,20,22,23,26]. Four of seven guidelines proposed other molecules at the first line as well (ceftriaxone, amoxicillin or cefuroxime-axetil) [18,19,20,26]. The IDSA guidelines were the only ones to suggest cefuroxime-axetil at the first line, and the ESGBOR guidelines were the only other ones to recommend it in the second line [18,19]. All the guidelines except the Polish ones recommended ceftriaxone at the second line [7,18,19,22,23,26]. The IDSA, ESGBOR and Belgium guidelines recommended amoxicillin at the first line on the same level as doxycycline [18,19,26].

The recommended duration was 28 days in all the guidelines for all the molecules except for ceftriaxone for which the IDSA, ESGBOR and Belgium guidelines suggested 14 to 28 days [18,19,26]. Seven guidelines recommended a second line of antibiotic therapy in case of failure after one month and suggested trying another molecule than the first [7,18,19,20,22,23,26].

All guidelines were homogeneous about the doses of all the antibiotics (doxycycline 200 mg/d, ceftriaxone 2 g/d, cefuroxime-axetil 500 mg × 2/d) except for amoxicillin for which the Belgium and IDSA guidelines recommended 500 mg × 3/d instead of 1 g × 3/d [19,26]. The rationale for recommending 1 g × 3/d is the included study using probenecid to increase the concentration of amoxicillin [62].

To conclude, these seven guidelines recommend doxycycline 200 mg/d at the first line for 28 days and ceftriaxone 2 g/d at the second line from 14 to 28 days, but with moderate to low evidence. In case of failure of the first line, another month of antibiotic therapy was recommended, changing the molecule. The differences between the guidelines can be explained by the absence of randomized controlled trials comparing the efficacy of the molecules between them and their optimal durations.

###### Children and Pregnant Women

Children were mentioned in 5/7 guidelines with a low level of evidence [7,18,22,23,26]. For children under eight years old, amoxicillin was the first-line therapy for 28 days: 30 mg/kg × 3/d in NICE guidelines [23], 80 mg/kg/d [7] or 100 mg/kg/d in 3 divided doses in the French guidelines [22] and 50 mg/kg/d in 3 divided doses in the Belgium guidelines [26]. The second line was ceftriaxone 80 mg/kg/d for 28 days [7,22,23]. The Belgium guidelines suggested ceftriaxone 100 mg/kg/d (max 2 g/d) at the first line as well, at the same level as amoxicillin [26]. For children over nine years old, doxycycline was recommended as first-line therapy: 4 mg/kg/j (max 200 mg/d] [7,22,26] or 5 mg/kg/j on day 1 then 2.5 mg/kg/d for 28 days [23]. In the French guidelines, ceftriaxone 2 g/d for 28 days was the first-line therapy for pregnant women [7,22]. The ESGBOR guidelines only mentioned that doxycycline should be avoided in children under eight years old and in pregnant or breastfeeding women [18]. The IDSA guidelines made no difference for children, adults and pregnant women because data are reassuring on the safety of short courses of doxycycline in these populations [19].

#### 2.2.4. Treatment of Other Manifestations (cf. Table 5)

##### LB Carditis (LC)

Lyme carditis is a rare manifestation of early disseminated infection with *B. burgdorferi* (0.3% to 5% of clinical manifestations) [7,19,20]. Thus, data regarding treatment are rare or nonexistent, and recommendations are based on an extrapolation of the treatment of other manifestations of LB (weak recommendation, low-quality evidence, grade AE) [7,19,23].

The most typical presentation is varying degrees of atrioventricular block (AVB) (first, second or third degree) [7,18,19,20]. Other conduction system abnormalities may be seen as well as atrial and ventricular arrhythmias. *B. burgdorferi* infection may also present as pericarditis, acute myocarditis or pancarditis [19,22].

Among the 12 selected recommendations, 7 dealt with LC [7,18,19,20,22,23,26] (Table 5). For patients with severe manifestations (complete AVB, advanced forms of partial heart block, including second-degree block or first-degree block with a PR interval of >0.3 s, syncope, unstable hemodynamic) and/or patients who require hospitalization, guidelines were consistent and recommended ceftriaxone 2 g/day for 14–21 days (6/7 guidelines) [7,18,19,22,23,26]. The two French guidelines, the ESGBOR guidelines and the American guidelines from IDSA proposed a switch to oral antibiotics as soon as clinical improvement is achieved [7,18,19,22]. Overall, patients with nonsevere manifestations of LC and/or patients who do not need to be hospitalized may be treated with doxycycline 100 mg × 2/day or 200 mg/day or amoxicillin 1.5 × 3d/day. Only the Belgium and NICE guidelines did not recommend amoxicillin for LC [23,26]. Outpatients may also be treated with cefuroxime-axetil according to ESGBOR guidelines and the American guidelines from IDSA [18,19]. The NICE guidelines proposed ceftriaxone 2g/d as an alternative treatment for these patients [23]. It was recommended to treat patients for a total of 14–21 days (6/7 guidelines) [7,18,19,22,23,26]. The Polish guidelines were less precise and proposed treatment using doxycycline, amoxicillin or ceftriaxone, regardless of cardiac manifestations, for 28–30 days (longer than the other guidelines) with no more explanation [20]. The NICE guidelines advised that azithromycin should not be used to treat people with cardiac abnormalities because of its effect on the QT interval [23].

For pregnant women and children under eight years old, Belgium and French High Health Authority guidelines recommend against using doxycycline (only second and third semesters for French guidelines) whereas American guidelines from IDSA indicated increasing favorable information on the safety of short courses of doxycycline in young children, which should impact the risk-to-benefit ratio of using this antibiotic in patients with various manifestations of LB [19,22,26].

##### Ophthalmological LB (OLB)

Among the 12 guidelines, only the two French guidelines mentioned the ophthalmological manifestations of LB [7,22]. This is not surprising because they are rare and poorly known. OLB may affect all anatomical structures of the eyes, the most frequent being uveitis and optic neuropathy. Other possible manifestations are early-stage conjunctivitis, keratitis, retinopathy and episcleritis [7,22].

There was no consensual treatment, and an ophthalmologist consultation was essential [22]. However, the two French guidelines recommended the same treatment as for LNB, i.e., ceftriaxone 2 g/d for 21 days (first-line treatment, grade AE) [7,22]. Because doxycycline has poor intraocular penetration, it is an option only for lesions localized on the surface of the eyes (except for keratitis) for 14 days [7]. Ceftriaxone is the preferred regimen for the treatment of keratitis and intraocular, orbital, or neuro-ophthalmological presentations (grade AE) [7]. Adjuvant corticosteroid therapy may be prescribed in some cases [7,22].

#### 2.2.5. Management of Post-Treatment Lyme Disease Syndrome (PTLDS) (cf. Table 5)

Post-treatment Lyme disease syndrome (PTLDS) refers to a pattern of nonspecific symptoms (fatigue, neurocognitive deficits, arthralgia or myalgia) that persist for more than six months after a proven and appropriately treated LB and which are caused neither by active nor persistent *B. burgdorferi* infection nor by other diseases [17,18,26]. In parallel, the concept of persistent manifestations or symptoms of LB following recommended treatment [7,19,27] or after a possible tick bite [22] was identified in other guidelines without a precise definition. 

Ten guidelines detailed the recommendations for ongoing symptoms [7,9,10,11,14,15,16,17,18,19]. The Swiss guidelines were exclusively focused on PTLDS and were based on a systematic review of the literature [9]. All the guidelines agreed to retreat in case of proven reinfection from another tick bite and to look for differential diagnoses in case of an absence of improvement after a first line of antibiotics for LB. The NICE guidelines and the French Scientific societies guidelines insisted on the necessity to clearly explain to the patient the possibility of continuing symptoms or sequelae without relation with an active LB, of symptoms of LB that may take months or years to resolve even after treatment and of alternative diagnoses that may explain their symptoms [7,15].

Five randomized, double-blind controlled studies showed that prolonged antibiotic therapy (>3 months) had no sustained benefit or even caused serious adverse effects [9,10,11,12]; however, the absence of well-conducted comparative trials to demonstrate the most efficient antibiotic therapy (molecule, dose, duration) for each manifestation of LB may lead to different interpretations and propositions to manage ongoing symptoms. 

Seven guidelines (including all those who specifically mentioned PTLDS) did not recommend antibiotic therapy and proposed symptomatic therapy (management of pain, cognitive symptoms, fatigue, etc.) and nonpharmacological approaches (counselling, regular low-impact aerobic exercise programs and cognitive behavioral therapy) [7,9,10,11,16,17,18]. The French scientific societies and the IDSA guidelines stated that patients should not receive additional, repeated or prolonged courses of antibiotics for persistent symptoms [7,19], as it is associated with adverse events such as allergic reaction, intravenous catheter complication and diarrhea and the selection of resistant bacteria. Antibiotic retreatment does not appear safe or effective and is discouraged [7,19]. Three guidelines suggested a possibility of treatment, after having eliminated all the differential diagnoses, in case of ongoing symptoms [14,15,19]. ILADS guidelines recommended oral antibiotics solely or in combination with injectable penicillin G benzathine or ceftriaxone alone or in combination with other antibiotics for 4–6 weeks for persistent symptoms after a tick bite or after a first line of antibiotics for LB [19]. They were the only ones to advocate for a prolonged antibiotic treatment based on an expert agreement with no scientific rationale [27]. The French High Health Authority suggested doxycycline for 28 days for persistent symptoms after a tick bite or after a first line of antibiotics for LB [14]. The NICE guidelines suggested a second line with another molecule in case of ongoing symptoms after a first line of antibiotics for LB [15]. 

## 3. Materials and Methods

### 3.1. Data Sources and Search Strategy

We performed a systematic review of guidelines on LB using the PRISMA method [68]. We searched MEDLINE, Google Scholar and databases of the National Institute for Health and Care Excellence (NICE, https://www.nice.org.uk/ accessed on 30 April 2021), The Arbeitsgemeinschaft der Wissenschaftlichen Medizinischen Fachgesellschaften (AWMF, https://www.awmf.org/ accessed on 30 April 2021), Haute Autorité de Santé (HAS, https://www.has-sante.fr/ accessed on 30 April 2021) and the International Lyme and Associated Diseases Society (ILADS, https://www.ilads.org/ accessed on 30 April 2021). We included guidelines from 2014 to 2020, as the majority of the guidelines have been updated recently. Only the most recent guidelines from the same authors or organizations were included in the analysis. Guidelines that only dealt with diagnosis were excluded from the analysis. Language was restricted to French, English, and German. The database search strategy combined the following terms: ‘guidelines’ or ‘recommendations’ and “Lyme disease” or “Lyme borreliosis” and “treatment” or “antibiotics” or “antibiotic therapy”. The systematic search was independently performed by AR and SN.

### 3.2. Assessment of the Quality of the Guidelines

To assess the methodological quality of each guidelines, the Appraisal of Guidelines for Research and Evaluation II (AGREE II) instrument was used [69] to evaluate 6 domains via 23 items: scope and purpose (domain 1), stakeholder involvement (domain 2), rigor of development (domain 3), clarity of presentation (domain 4), applicability (domain 5) and editorial independence (domain 6). We rated each item on a seven-point scale, with 1 being the lowest rating and 7 being the highest. Two reviewers (AR and SN) independently assessed the quality of each guideline that met the eligibility criteria. If the detailed methodology of the guidelines was accessible in a supplementary document, it was considered during the AGREE II assessment. In the event of disagreement, a third author (JS) was asked to evaluate every item for which the rating differed by more than 1 point, and we achieved consensus through discussion. After consensus meeting, inter-rater reliability was measured using Cohen’s weighted kappa coefficient, and statistical comparisons between scores were performed using two-sided *t*-test. The domain score was calculated as described in AGREE II by the formula: (obtained score-minimum possible score)/(maximum possible score-minimum possible score) [59,69,70].

The maximum possible score was maximum possible score for each item × number of items in domain × number of appraisers. The minimum possible score was minimum possible score for each item × number of items in domain × number of appraisers. We considered domain scores <50% as low quality [59,69,70].

### 3.3. Data Extraction of Guidelines and Analysis

After quality appraisal, AR and SN independently extracted information and checked the data from each guideline. A standard form was designed for data extraction covering the country, publication year, level of scientific evidence, clinical stage of LB and antibiotic therapy (molecule, duration, and dosage). Descriptive analysis of the quality of the guideline included median and interquartile range (IQR). STATA version IC 16 (College Station, TX, USA) was used for all statistical analyses. 

No ethical approval or consent was needed for this study.

## 4. Conclusions

European and American guidelines were quite homogeneous regarding the recommended molecules (mostly oral doxycycline as preferential treatment in all situations, and intravenous ceftriaxone as the main alternative), their durations varying from 10 to 28 days and their dosage. The main differences were due to the lack of well-conducted comparative trials, making these guidelines rely on moderate to low level of evidence. A successful treatment was defined in most guidelines by the resolution of the symptoms. In case of persistence of symptoms, all the guidelines recommended additional explorations to search the differential diagnoses. The ILADS guidelines were the only ones to suggest longer duration and additional antibiotics for more than 2 months for prolonged symptoms based on an expert consensus. Well-conducted comparative trials such as randomized controlled trials are needed to assess the best molecules, the optimal duration and the most effective doses.

## Figures and Tables

**Figure 1 pathogens-10-00972-f001:**
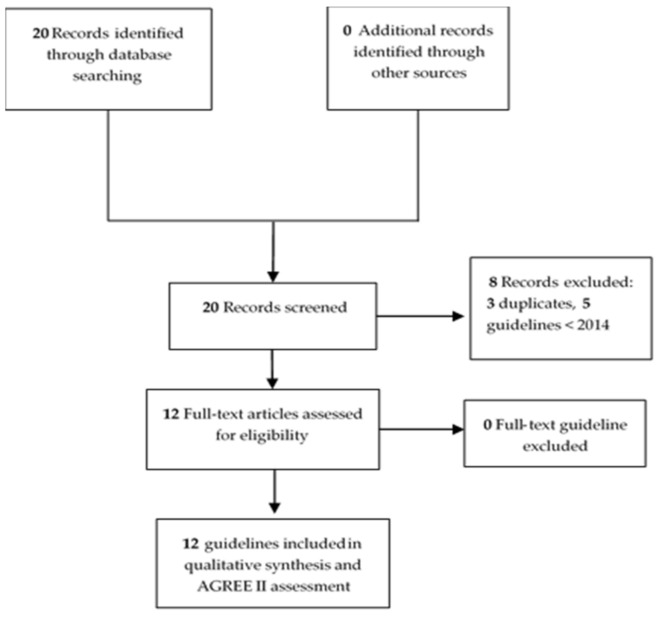
Study flow diagram.

**Table 1 pathogens-10-00972-t001:** Characteristics of the 12 selected guidelines.

	Guidelines	Country	Institution	Year	Type
1	ILADS guidelines [27]	USA	ILADS	2014	Evidence-based and expert consensus
2	Polish guidelines [20]	Polish	PSEID	2015	Evidence-based
3	Swiss guidelines [17]	Swiss	SSID and SSN	2016	Evidence-based
4	Belgium guidelines [26]	Belgium	BAPCOC	2016	Evidence-based
5	German Dermatology Society guidelines for cutaneous LB [24]	Germany	AWMF	2017	Evidence-based
6	ESGBOR guidelines [18]	Europe	ESGBOR	2017	Evidence-based
7	French High Health Authority guidelines [22]	France	HAS	2018	Evidence-based
8	NICE guidelines [23]	England	NICE	2018	Evidence-based
9	German Neurology Society guidelines for LNB [25]	Germany	AWMF	2019	Evidence-based
10	French scientific societies guidelines [7]	France	French scientific societies	2019	Evidence-based
11	Canadian guidelines, Prince Edward Island [21]	Canada	Department of Health	2019	Evidence-based
12	American guidelines [19]	USA	IDSA	2020	Evidence-based

ILADS = International Lyme and Associated Diseases Society; PSEID = the Polish Society of Epidemiology and Infectious diseases, SSID = Swiss Society of Infectious Disease, SSN = Swiss Society of Neurology; BAPCOC = Belgian Antibiotic Policy Coordination Committee; AWMF = The Arbeitsgemeinschaft der Wissenschaftlichen Medizinischen Fachgesellschaften; ESGBOR = European Study Group for Lyme Borreliosis; HAS = Haute Autorité de Santé, High Health Authority; NICE = National Institute for Health and Care Excellence; IDSA = Infectious Diseases Society of America.

**Table 2 pathogens-10-00972-t002:** AGREE II domain scores for the 12 guidelines, after a blind evaluation by two appraisers.

Guidelines	Scope and Purpose	Stakeholder Involvement	Rigor of Development	Clarity of Presentation	Applicability	Editorial Independence	Overall Assessment	Recommended
ILADS [27]	100%	69%	58%	22%	31%	13%	42%	*N*
PSEID [20]	81%	31%	10%	58%	44%	29%	33%	*N*
SSID and SSN [17]	100%	58%	89%	67%	81%	67%	83%	*Y*
BAPCOC [26]	72%	50%	40%	67%	71%	58%	53%	*YM*
AWMF [24]	100%	100%	96%	86%	81%	100%	100%	*Y*
ESGBOR [18]	75%	58%	66%	42%	44%	79%	50%	*YM*
HAS [22]	100%	100%	88%	83%	79%	54%	75%	*YM*
NICE [23]	100%	100%	94%	89%	100%	100%	100%	*Y*
AWMF [25]	100%	100%	98%	89%	94%	100%	100%	*Y*
French scientific societies [7]	100%	100%	99%	87%	92%	100%	100%	*Y*
Canada [21]	64%	36%	17%	39%	85%	42%	25%	*N*
IDSA [19]	100%	100%	100%	63%	94%	96%	100%	*Y*
Median ± IQR	100% ± 22%	85% ± 46%	88% ± 48%	67% ± 35%	81% ± 36%	73% ± 52%	79% ± 54%	
**Cohen’s kappa**	-	-	-	-	-	-	0.90 (0.84–0.96)	

Y = yes, YM = yes with modifications, N = no, ILADS = International Lyme and Associated Diseases Society; PSEID = the Polish Society of Epidemiology and Infectious diseases, SSID = Swiss Society of Infectious Disease, SSN = Swiss Society of Neurology; BAPCOC = Belgian Antibiotic Policy Coordination Committee; AWMF = The Arbeitsgemeinschaft der Wissenschaftlichen Medizinischen Fachgesellschaften; ESGBOR = European Study Group for Lyme Borreliosis; HAS = Haute Autorité de Santé, High Health Authority; NICE = National Institute for Health and Care Excellence; IDSA = Infectious Diseases Society of America.

**Table 3 pathogens-10-00972-t003:** Comparative table of the treatment recommended in the European and American guidelines for cutaneous LB.

Guidelines	Erythema Migrans		Borrelial Lymphocytoma		Acrodermatitis Chronica Atrophicans	
	First Intention	Second Intention	First Intention	Second Intention	First Intention	Second Intention
**ILADS** [27]	Amoxicillin (1.5–2.0 g/d) or Cefuroxime-axetil (500 mg × 2/d) or Doxycycline (100 mg × 2/d) for 4–6 weeks or Azithromycin (250–500 mg/d) for 21 days minimum **level of evidence = very low**		NA	NA	NA	NA
**PSEID** [20]	Doxycycline (100 mg × 2/d), for 14–28 days **level of evidence = NA**	Amoxicillin (1.5–2 g/d) or Cefuroxime-axetil (500 mg × 2/d),for 14-28 days **level of evidence = NA**	Doxycycline (100 mg × 2/d) or Amoxicillin (1.5–2 g/d) or Cefuroxime axetil 500 mg × 2/d for 14–28 days **level of evidence = NA**		Doxycycline (100 mg × 2/d) or Ceftriaxone (2 g/d) or Amoxicillin (1.5–2 g/d) or Cefuroxim-axetil (500 mg × 2/d) for 14-21 days **level of evidence = NA**	
**SSID and SSN** [17]	NA	NA	NA	NA	NA	NA
**BAPCOC** [26]	Doxycycline (100 mg × 2/d) for 10 days **level of evidence = NA**	Amoxicillin (1.5–2 g/d) or Cefuroxime-axetil (500 mg × 2/d), or Clarythromycin (500 mg × 2/d) for 14 days or Azithromycin (1 g on D1, then 500 mg/d) for 4 days or (500 mg/d) for 7 days **level of evidence = NA**	NA	NA	Doxycycline (100 mg × 2/d) for 21–28 days or Ceftriaxone (2 g/d) for 14 days **level of evidence = NA**	
**AWMF** [24]	Doxycycline (100 mg × 2/d or 200 mg/d) for 10–14 days or Amoxicillin (0.5–1 g × 3/d) or Cefuroxime axetil (500 mg × 2/d) for 14 days or Azithromycin (250 mg × 2/d) for 5–10 days **level of evidence = NA**		Doxycycline (100 mg × 2/d or 200 mg/d) or Amoxicillin (0.5–1 g × 3/d) or Cefuroxime-axetil (500 mg × 2/d), for 14–21 days or Azithromycin (250 mg × 2/d) for 5–10 days **level of evidence = NA**		Doxycycline (100 mg × 2/d or 200 mg/d) or Amoxicillin (0.5–1 g × 3/d) for 30 days If associated with neurological symptoms: Penicillin G (4 × 5MU/d) or Ceftriaxone (2 g/d) or Cefotaxime (2 g × 3/d) for 14–21 days **level of evidence = NA**	
**ESGBOR** [18]	Doxycycline for 10 days or Amoxicillin or Cefuroxime-axetil or Phenoxymethylpenicillin for 14 days **level of evidence = NA**	Azithromycin for 5–10 days, if CI for β-lactams or tetracyclines **level of evidence = NA**	Doxycycline or Amoxicillin or Cefuroxime-axetil or Phenoxymethylpenicillin for 14 days **level of evidence = NA**	Azithromycin for 5–10 days, if CI for β-lactams or tetracyclines **level of evidence = NA**	Doxycycline or Amoxicillin or Cefuroxime-axetil or Phenoxymethylpenicillin for 21–28 days **level of evidence = NA**	Azithromycin for 5–10 days, if CI for β-lactams or tetracyclins **level of evidence = NA**
**HAS** [22]	Doxycycline (100 mg × 2/d or 200 mg/d) or Amoxicillin (1 g × 3/d), for 14 days **level of evidence = moderate**	Azithromycin (1 g on D1, then 500 mg/d) for 7 days **level of evidence = moderate**	Doxycycline (100 mg × 2/d or 200 mg/d) or Amoxicillin (1–2 g × 3/d), for 21 days **level of evidence = low**	Azithromycin (1 g on D1, then 500 mg/d) for 10 days **level of evidence = low**	Doxycycline (200 mg/d) for 28 days **level of evidence = low**	Ceftriaxone (2 g/d), for 28 days **level of evidence = low**
**NICE** [23]	Doxycycline (100 mg × 2/d or 200 mg/d) for 21 days **level of evidence = low**	Amoxicillin (1 g × 3/d) for 21 days or Azithromycin (500 mg/d) for 17 days **level of evidence = low**	NA	NA	Doxycycline (100 mg × 2/d or 200 mg/d) for 28 days **level of evidence = low**	Amoxicillin (1 g × 3/d) or Ceftriaxone (2 g/d), for 28 days **level of evidence = low**
**AWMF** [25]	NA	NA	NA	NA	NA	NA
**French scientific societies** [7]	Doxycycline (100 mg × 2/d) for 14 days **level of evidence = grade B**	Amoxicillin (1 g × 3/d) for 14 days **level of evidence = grade B**	Doxycycline (100 mg × 2/d) for 21 days **level of evidence = grade B**	Amoxicillin (1 g × 3/d) for 21 days **level of evidence = grade B**	Doxycycline (200 mg/d) for 28 days **level of evidence = grade B**	Ceftriaxone (2 g/d) for 28 days **level of evidence = grade B**
**Canada** [21]	Doxycycline (100 mg × 2/d) for 14 days **level of evidence = NA**	Amoxicillin (0.5 g × 3/d) or Cefuroxime-axetil (500 mg × 2/d) for penicillin-allergic patient, for 14 days **level of evidence = NA**	NA	NA	NA	NA
**IDSA** [19]	Doxycycline (100 mg × 2/d or 200 mg/d) for 10 days or Amoxicillin (0.5 g × 3/d) or Cefuroxime-axetil (500 mg × 2/d) for 14 days **level of evidence = moderate**	Azithromycin (500 mg/d) for 5–10 days **level of evidence = moderate**	Doxycycline (100 mg × 2/d or 200 mg/d) for 21 days or Amoxicillin (0.5–1 g × 3/d) for 14 days or Cefuroxime-axetil (500 mg × 2/d) for 14–21 days **level of evidence = low**		Doxycycline (100 mg × 2/d or 200 mg/d) or Amoxicillin (0.5–1 g × 3/d) or Cefuroxime-axetil (500 mg × 2/d) for 21–28 days **level of evidence = low**	

ILADS = International Lyme and Associated Diseases Society; PSEID = the Polish Society of Epidemiology and Infectious Diseases, SSID = Swiss Society of Infectious Disease, SSN = Swiss Society of Neurology; BAPCOC = Belgian Antibiotic Policy Coordination Committee; AWMF = The Arbeitsgemeinschaft der Wissenschaftlichen Medizinischen Fachgesellschaften; ESGBOR = European Study Group for Lyme Borreliosis; HAS = Haute Autorité de Santé, High Health Authority; NICE = National Institute for Health and Care Excellence; IDSA = Infectious Diseases Society of America; NA = Not available; EA = Expert Agreement; CI = contra-indication.

**Table 4 pathogens-10-00972-t004:** Comparative table of the treatment recommended in the European and American guidelines for Lyme neuroborreliosis and Lyme arthritis.

Guidelines	Early Lyme Neuroborreliosis		Late Lyme Neuroborreliosis		Early (<6 Months) and Late (>6 Months) Lyme Arthritis	
	First Intention	Second Intention	First Intention	Second Intention	First Intention	Second Intention
**ILADS** [27]	NA	NA	NA	NA	NA	NA
**PSEID** [20]	Cranial nerves deficit: Doxycycline 100 mg × 2/d for 14–28 days Meningitis, radiculitis, vasculitis: Doxycycline (100 mg × 2/d) or Ceftriaxone (2g/d) for 14–28 days **level of evidence = NA**		Encephalomyelitis, radiculoneuritis, meningitis, occlusive vasculitis stroke: Ceftriaxone (2 g/d) for 21–28 days. **level of evidence = NA**		Doxycycline (100 mg × 2/d) Or Ceftriaxone (2 g/d) for 28–30 days **level of evidence = NA**	If failure: continue the antibiotics one more month **level of evidence = NA**
**SSID and SSN** [17]	NA	NA	NA	NA	NA	NA
**BAPCOC** [26]	Doxycycline (100 mg × 2/d) or Ceftriaxone (2 g/d) for 14 days **level of evidence = NA**		Ceftriaxone (2 g/d), for 28 days **level of evidence = NA**		Doxycycline (100 mg × 2/d) or Amoxicillin (500 mg × 3/d), for 28 days **level of evidence = NA**	If failure: Ceftriaxone (2 g/d) for 14–28 days or another oral line for 28 days **level of evidence = NA**
**AWMF** [24]	NA	NA	NA	NA	NA	NA
**ESGBOR** [18]	For ambulatory patient: Doxycycline (100 mg × 2/d) for 14 days For hospitalized patients: Ceftriaxone (2 g/d) for 14 days **level of evidence = NA**		Encephalomyelitis: Ceftriaxone (2 g/d) for 14–28 days **level of evidence = NA**		Doxycycline or Amoxicillin for 28 days **level of evidence = NA**	If failure: Ceftriaxone for 14–28 days Or Cefuroxime-axetil for 28 days **level of evidence = NA**
**HAS** [22]	Ceftriaxone (2 g/d) or Doxycycline (100 mg × 2/d) for 21 days Children with isolated nerve palsy: Amoxicillin (100 mg/kg/d × 3/d) for 21 days **level of evidence = NA**		Ceftriaxone (2 g/d) (100 mg/kg/d) for 28 days **level of evidence = NA**	Doxycycline (100 mg × 2/d or 4 mg/kg/d) or Penicillin G (24 MU/d) for 28 days **level of evidence = NA**	Doxycycline (200 mg/d) for 28 days **level of evidence = low**	If failure: Ceftriaxone (2 g/d) for 28 days **level of evidence = low**
**NICE** [23]	PNS: Doxycycline (100 mg × 2/d or 200 mg/d) for 21 days CNS: Ceftriaxone (2 g × 2/d or 4 g/d), for 21 days **level of evidence = moderate/very low**	PNS: Amoxicillin (1 g × 3/d) for 21 days CNS: Doxycycline (200 mg × 2/d or 400 mg/d) for 21 days **level of evidence = moderate/ very low**	NA	NA	Doxycycline (100 mg × 2/d or 200 mg/d) for 28 days **level of evidence = low /very low**	1st Alternative or failure: Amoxicillin (1 g × 3/d) for 28 days 2nd alternative or failure: Ceftriaxone (2 g/d) for 28 days **level of evidence = low/very low**
**AWMF** [25]	Doxycycline (100 mg × 2/d or 100 mg × 3/d or 200–300 mg/d) or Ceftriaxone (2 g/d) or Penicillin G (5 MU/d) for 14 days **level of evidence = grade Ia**		Ceftriaxone (2 g/d) or Cefotaxime (2 g × 3/d) or Penicillin G (5 MU/d) or Doxycycline (100 mg × 2/d or 100 mg × 3/d or 200–300mg/d) for 14–21 days **level of evidence = grade Ia to grade III**		NA	NA
French scientific societies [7]	Doxycycline (100 mg × 2/d) for 14 days **level of evidence = grade EA**	Ceftriaxone (2 g/d) for 14 days **level of evidence = grade EA**	PNS: Doxycycline (100 mg × 2/d) for 21 days CNS: Doxycycline (200 mg × 2/d) for 21 days **level of evidence = grade EA**	Ceftriaxone (2 g/d) for 21 days **level of evidence = grade EA**	Doxycycline (200 mg/d) for 28 days **level of evidence = grade EA**	If failure: Ceftriaxone (2 g/d) for 28 days **level of evidence = grade EA**
**Canada** [21]	NA	NA	NA	NA	NA	NA
**IDSA** [19]	Meningitidis or radiculitis: Doxycycline (100 mg × 2/d or 200mg/d) or Ceftriaxone (2 g/d) for 14–21 days Cranial nerve palsy: Doxycycline (100 mg × 2/d or 200mg/d) for 14–21 days Parenchymal involvement of the brain or spinal cord: Ceftriaxone (2 g/d) or Cefotaxime (2 g × 3/d) or Penicillin G (18–24 MU) for 14–21 days **level of evidence = strong recommendation moderate-quality evidence**	Meningitidis or radiculitis: Cefotaxime (2 g × 3/d) or Penicillin G (18–24 MU) for 14-21 days **level of evidence = strong recommendation moderate-quality evidence**	NA	NA	Doxycycline (200 mg/d or 100mg × 2/d) or Amoxicillin (500 mg × 3/d) or Cefuroxim axetil (500 mg × 2/d) for 28 days **level of evidence = strong recommendation moderate-quality evidence**	If partial response (mild residual joint swelling): no antibiotic, search for differential diagnosis, and then eventually discuss a 2nd line of oral antibiotics **level of evidence = strong recommendation moderate-quality evidence**

ILADS = International Lyme and Associated Diseases Society; PSEID = the Polish Society of Epidemiology and Infectious diseases, SSID = Swiss Society of Infectious Disease, SSN = Swiss Society of Neurology; BAPCOC = Belgian Antibiotic Policy Coordination Committee; AWMF = The Arbeitsgemeinschaft der Wissenschaftlichen Medizinischen Fachgesellschaften; ESGBOR = European Study Group for Lyme Borreliosis; HAS = Haute Autorité de Santé, High Health Authority; NICE = National Institute for Health and Care Excellence; IDSA = Infectious Diseases Society of America; NA = Not available; CNS = Central Nervous System; PNS = Peripheral Nervous System; EA = Expert Agreement.

**Table 5 pathogens-10-00972-t005:** Comparative table of the treatment recommended in the European and American guidelines for Lyme carditis, ophtalmological Lyme borreliosis and Post-Treatment Lyme Disease Syndrome.

Guidelines	Lyme Carditis		Ophtalmological Lyme Borreliosis		Post-Treatment Lyme Disease Syndrome	
	First Intention	Second Intention	First Intention	Second Intention	First Intention	Second Intention
**ILADS** [27]	NA		NA	NA	If persistent symptoms, treatment options are extensive and choices must be individualized: oral antibiotics alone or in combination or injectable penicillin G benzathine or ceftriaxone alone or in combination with other antibiotics for 4–6 weeks level of evidence = low	
**PSEID** [20]	Doxycycline (100 mg × 2/d) or Amoxicillin (1.5–2g/d) or Ceftriaxone (2 g/d) for 28–30 days **level of evidence = NA**		NA	NA	NA	NA
**SSID and SSN** [17]	NA	NA	NA	NA	Antibiotic retreatment is not recommended after appropriate initial antibiotics for LB. No evidence for specific treatment. **level of evidence = strong**	
**BAPCOC** [26]	Doxycycline (100 mg × 2/d) for 21 days Or Ceftriaxone (2 g/d) for 14 days (preferred in more severe cases) **level of evidence = NA**		NA	NA	Antibiotic therapy is not recommended **level of evidence = NA**	
**AWMF** [24]	NA	NA	NA	NA	Benefit of repeated and long-term antibiotics not verified. **Level of evidence = NA**	
**ESGBOR** [18]	Outpatients: Doxycycline or Amoxicillin or Cefuroxime-axetil Hospitalized patients: Ceftriaxone for 14–21 days A switch to oral antibiotic can be made if improvement **level of evidence = NA**		NA	NA	Antibiotic therapy is not recommended **level of evidence = NA**	
**HAS** [22]	Outpatients: Doxycycline (200 mg/d) or Amoxicillin (3 g/d) Hospitalized patients: Ceftriaxone (2 g/d) with a switch as soon as possible to oral antibiotics for 21 days **level of evidence = NA**		Ceftriaxone (2 g/d or 100 mg/kg/d) for 28 days **level of evidence = NA**		After having eliminated differential diagnosis to LB, consider: Doxycycline (200 mg/d) for 28 days **level of evidence = NA**	If allergy to doxycycline: Azithromycin (1 g on D1, then then 500 mg/d) for 15 days. **level of evidence = NA**
**NICE** [23]	Stable patients: Doxycycline (100 mg 2 × /d or 200 mg/d) Hemodynamically unstable patients: Ceftriaxone (2 g/d) for 21 days **level of evidence = grade EA**	Stable patients: Ceftriaxone (2 g/d) for 21 days **level of evidence = grade EA**	NA	NA	Consider a second course of antibiotics for people with ongoing symptoms if treatment may have failed. Use an alternative antibiotic to the initial course. If a person has ongoing symptoms following 2 completed courses of antibiotics for LB: do not routinely offer further antibiotics and consider discussion with a national reference laboratory or discussion or referral to a specialist **level of evidence = EA**	
**AWMF** [25]	NA	NA	NA	NA	Patients should not be treated with antibiotics. **level of evidence = NA**	
**French scientific societies** [7]	Patient with syncope, type 2 or 3 AVB, or type 1 AVB > 30 ms: Ceftriaxone (2 g/d) with a switch to oral antibiotics as soon as continuous cardiac monitoring is no longer required Patients with other manifestations: Doxycycline (100 mg × 2/d) or Amoxicillin (1 g × 3/d) for a total of 21 days **level of evidence = grade C to EA**		Lesions on the surface of the eyes: Doxycycline (200 mg/d) or Ceftriaxone (2 g/d) for 14 days Keratitis, intraocular, orbital, neuro-ophthalmological lesions: Ceftriaxone (2 g/d) for 21 days **level of evidence = NA**	Doxycycline (100 mg × 2/d) for 21 days **level of evidence = NA**	Patients should not receive repeated or prolonged courses of antibiotics. **level of evidence = grade A**	
**Canada** [21]	NA	NA	NA	NA	NA	NA
**IDSA** [19]	Outpatients: Doxycycline (100 mg × 2/d or 200 mg/d) or Amoxicillin (500 mg × 3/d) or Cefuroxime axetil (500 mg × 2/d) Hospitalized patients: initially Ceftriaxone (2 g/d) then switching to oral antibiotics For a total of 14–21 days **Level of evidence = weak recommendation, very** **low-quality evidence**		NA	NA	Additional antibiotic therapy is not recommended. **Level of evidence = strong recommendation, moderate-quality evidence.**	

ILADS = International Lyme and Associated Diseases Society; PSEID = the Polish Society of Epidemiology and Infectious diseases, SSID = Swiss Society of Infectious Disease, SSN = Swiss Society of Neurology; BAPCOC = Belgian Antibiotic Policy Coordination Committee; AWMF = The Arbeitsgemeinschaft der Wissenschaftlichen Medizinischen Fachgesellschaften; ESGBOR = European Study Group for Lyme Borreliosis; HAS = Haute Autorité de Santé, High Health Authority; NICE = National Institute for Health and Care Excellence; IDSA = Infectious Diseases Society of America; NA = Not available; EA = Expert Agreement.

## Data Availability

All the data of the statistical analysis are available by email at alice.raffetin@chiv.fr or steve.nguala@chiv.fr.

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
