# Peer review of "Methodological Quality Assessment with the AGREE II Scale and a Comparison of European and American Guidelines for the Treatment of Lyme Borreliosis: A Systematic Review"

_pathogens, 2021, doi:10.3390/pathogens10080972_

Round 1

Reviewer 1 Report

Review on the manuscript ID: pathogens-1243055

Title: Methodological quality assessment with the AGREE II scale and a comparison of European and American guidelines for the treatment of Lyme borreliosis: a systematic review

by Nguala et al.

A complex and elaborate review on major recommendations regarding treatment of Lyme borreliosis.

General comment: especially the extensive and large tables (3, 4, 5) are difficult to read and it is questionable whether it is necessary to include all this information in upright format tables.

The Arbeitsge-meinschaft der Wissenschaftlichen Medizinischen Fachgesellschaftten should always read: The Arbeitsgemeinschaft der Wissenschaftlichen Medizinischen Fachgesellschaften (lines 75, 89, 137+138, 253, 417, 516, 525+526)

Lins 52: …compared to those in the USA (B. burgdorferi sensu strictu)(3).

phenoxymethylpenicillin should always be written in English (line 178, table 3 ESGBOR)

abbreviation EA in the tables (always included in the legend): no EA in table 3, AE in table 4 (is it EA?)

table 4: PSEID treatment recommendation for Lyme arthritis second line: If failure: another month? Another month of one of the two first line antibiotics?

Material and methods:

The process to involve a third author remains unclear: was it always the same third author? How often it was necessary to involve a third author? How large was the discrepancy (disagreement) to decide for involvement of a third author?

The overall assessment criteria is only described for recommendation or no recommendation – what is the criterion for recommendation with modification?

Line 561: interquartile range (IQR)

Reviewer 2 Report

This manuscript systematically compares 12 guidelines for the treatment of Lyme borreliosis. The review is useful, elaborate, clear and well written.

Some remarks:

  1. The introduction should present epidemiological and incidence data.
  2. The introduction should clarify that in vitro antibiotic resistance has not been observed. There is no objective evidence to indicate that antibiotic tolerant persister cells are clinically relevant. There is no evidence that posttreatment Lyme disease symptoms are caused by persistence of the pathogen. 
  3. Infections regarded as late (e.g. late LA) should be clearly defined by timing.
  4. Table 2. There are overall assessment scores of 100% that are based on <100% domain scores.
  5. In general, it is not clear whether the guidelines rely solely on resolution of symptoms and lack of relapse as the indication of a successful treatment, or are based additionally on laboratory or clinical diagnosis.
  6. The review indicates that the current guidelines are based on a low and unsatisfactory level of evidence. The authors should state which data is required. What is the meaning of "well-conducted comparative trials" that are mentioned in the discussion? Present suggestions.
  7. Tables 3-5 are difficult to read in the present form. Consider revising.
  8. Table 3. "Azithromycine" should be corrected to Azithromycin.
  9. Line 243 "ß-lactamsfor" should be corrected to "ß-lactams for".
  10. Some brackets are not preceded by a space (e.g. line 187).

Reviewer 3 Report

The manuscript by Nguala and colleagues entitled "Methodological quality assessment with the AGREE II scale and a comparison of European and American guidelines for the treatment of Lyme borreliosis: a systematic review" attempts to present a comprehensive review of 12 Lyme guidelines from Europe and the USA. Unfortunately the attempt is plagued by misinterpretation of selective data and mischaracterization of evidence that propagates myths about tick-borne diseases that are potentially damaging to patients with these diseases.

Major Comments

1. The Introduction sets the tone for the analysis: Lyme disease is clinically distinct in Europe versus the USA (so hard to compare diagnosis and treatment), laboratory testing is based on two-tier serology (which misses more than half of Lyme cases), antibiotic monotherapy is limited to 21-28 days (which may be inadequate for patients with prolonged infection) and "there is no scientific evidence" for prolonged Borrelia infection. This Introduction sets the analysis on the road to destruction of its credibility and generalizability.

2. Only three of the Lyme guidelines follow the GRADE process of evidence evaluation. Without this approach, the Institute of Medicine states that guidelines may not be trustworthy (Hayawi et al, PLoS One 2018;13(12):e0208251). The problem of COIs in guidelines development is not discussed in the analysis (Johnson & Stricker, PEHM 2010:5:9).

3. The Lyme guidelines are based on generally weak evidence, leaving interpretation of that evidence to the biases and whims of the guidelines authors. Although the Agree process may show agreement of these biases and whims, the resulting analysis is eminence-based rather than evidence-based. The process therefore favors time-honored myths about tick-borne diseases that are harmful to suffering patients while ignoring other approaches to the disease set out in the peer-reviewed literature (Stricker & Johnson, Infect Drug Resist 2016:9 215–219; Rebman & Aucott, Front Med 2020 ;7:57).

4. It is of interest that the ILADS guidelines are singled out as not being "evidence-based". These guidelines are based on 213 peer-reviewed references with careful analysis using the GRADE method. Therefore the ILADS guidelines have a higher standard of evidence than most of the other guidelines in the study. This misrepresentation reveals the bias of the authors of this screed.

5. Lyme disease has become a much more complex illness, with tick-borne coinfections often complicating the diagnosis and treatment of the disease. The lack of sensitive commercial testing coupled with weak clinical research evidence and limited treatment approaches creates a significant problem for any reliable guidelines analysis. Add to this the controversy over chronic Lyme disease, which has only recently been defined (Stricker & Fesler, Am J Infect Dis 2018;14:1-44) and continues to be ignored by most of the guidelines authors. The guidelines analysis does not apply to the vast majority of these patients, and publication of an inadequate analysis will only do more harm to patients who do not fit the procrustean myths in this analysis.

Minor Comments

1. Page 2, line 54: beta lactam.

2. References 8 and 65 are the same.

Round 2

Reviewer 3 Report

The authors have refused to correct the many myths and inaccuracies in their report, citing literature that is either biased and/or scientifically flawed. Two examples:

  1. They now categorize the GRADE-based ILADS guidelines as "evidence-based and expert consensus". This description is laughable when compared to all the other expert-consensus non-GRADE guidelines that the authors consider to be evidence-based. The critique of the ILADS guidelines in Table 3 is inaccurate.
  2. They state that coinfections are rare in Lyme disease patients. This statement is contradicted by extensive peer-reviewed literature, as noted in my previous review.

Eminence-based consensus is not science. My previous review is unchanged.

Author Response

Dear reviewer,

We have carefully read your comments and we thank you for the time spent to improve our work.  We have corrected the manuscript and we have sent our answers previously. Unfortunately, we cannot positively answer to the other suggestions and we will not be able to change our manuscript as it would be scientifically uncorrect. 

The suggested articles are promoting non validated theories about the diagnostic tests, the persistence of Borrelia, and prolonged antibiotic therapies. 

Moreover, the remarks about our methodology appear unfounded as we strictly followed the AGREE II tool by a double-blind evaluation. 

We still do not agree with the new remarks adressed to us as they do not correspond to the scientific evidence and to the actual scientific knowledge. We have cited the articles that were supporting our work, sharing their analysis with the best scientific societies in the world (IDSA, NICE, HAS etc.). 

Yours sincerely,

The authors